# Inflammation-Associated Stem Cells in Gastrointestinal Cancers: Their Utility as Prognostic Biomarkers and Therapeutic Targets

**DOI:** 10.3390/cancers16183134

**Published:** 2024-09-12

**Authors:** Beauty Kumari, Aniket Tiwari, Sakshi Meena, Dinesh Kumar Ahirwar

**Affiliations:** 1Department of Bioscience & Bioengineering, Indian Institute of Technology Jodhpur, Jodhpur 342030, Rajasthan, India; p22bb003@iitj.ac.in (B.K.); tiwari.21@iitj.ac.in (A.T.); 2School of Life Sciences, Devi Ahilya Vishwavidyalaya Indore, Indore 452001, Madhya Pradesh, India; sakshimeena9589@gmail.com

**Keywords:** inflammation, stem cells, cancer stem cells, gastrointestinal cancer, biomarkers

## Abstract

**Simple Summary:**

Stem cells are unique cell types in our body that play a key role in organ development and repair. Various inflammatory molecules act as critical regulators of stem cell activity. The dysregulation of inflammation and stem cell signaling pathways leads to a variety of diseases, including cancer. This review summarizes the role of inflammatory signals in regulating stem cell function within the gastrointestinal (GI) tract. We further discuss the studies highlighting the role of stem cells in the initiation and progression of cancers of the GI tract. We also report on various studies that assess the prognostic and therapeutic utility of cancer stem cells and their molecular markers in GI cancers.

**Abstract:**

Stem cells are critical for the development and homeostasis of the gastrointestinal (GI) tract. Inflammatory molecules are known to regulate the activity of stem cells. A comprehensive review specifically describing the role of inflammatory molecules in the regulation of stem cells within the GI tract and in GI cancers (GICs) is not available. This review focuses on understanding the role of inflammatory molecules and stem cells in maintaining homeostasis of the GI tract. We further discuss how inflammatory conditions contribute to the transformation of stem cells into tumor-initiating cells. We also describe the molecular mechanisms of inflammation and stem cell-driven progression and metastasis of GICs. Furthermore, we report on studies describing the prognostic value of cancer stem cells and the clinical trials evaluating their therapeutic utility. This review provides a detailed overview on the role of inflammatory molecules and stem cells in maintaining GI tract homeostasis and their implications for GI-related malignancies

## 1. Introduction

The gut is a highly specialized and complex organ of the body that helps in nutrient absorption and defense against infections. The gut lining, or the gastrointestinal (GI) tract lining, is constantly exposed to various detrimental agents, including environmental pollutants, food irritants, and infectious agents, that may compromise its integrity, making it vulnerable to chronic inflammation-induced diseases [1,2,3]. Among various mechanisms of GI tract repair, stem cells play a critical role in curbing inflammation and repairing inflammation-induced tissue damage. The maintenance of GI tract homeostasis relies on a delicate balance of stem cell proliferation, differentiation, the elimination of dysfunctional intestinal cells, and the integration of new cells into the epithelium [4]. Intestinal stem cells (ISCs) reside within the intestinal crypts and create a dynamic microenvironment where neutral drift dynamics ensure continual renewal [5,6,7,8]. ISCs progeny migrates upward from the base of the crypts towards the surface, differentiating into specialized cells like goblet cells and enterocytes, essential for regenerating the gut lining and combating inflammation, thereby sustaining overall gut health [9]. Repeated exposure to damaging agents, like food chemicals and infectious agents, activates inflammatory responses. If unresolved, these prolonged inflammatory activities and their associated molecules pose a significant risk of disrupted stem cell niche and, thereby, the initiation of a variety of inflammatory diseases, including cancers of the GI tract. Constantly changing lifestyles, food habits, and industrial development have increased the risk of cancers, including gastrointestinal cancer (GIC) [10]. Globally, 4.8 million new cases of GIC were reported in 2018, resulting in 3.4 million deaths. GI malignancies accounted for 35% of all cancer-related deaths and 26% of global cancer worldwide. Among them, colorectal, stomach, and intestinal cancer are the most prevalent type of GIC; therefore, we have focused this review on this subtypes of GIC [10]. These cancers share many common risk factors, including alcohol consumption, tobacco smoking, diet, and obesity [11,12,13]. Due to late-stage diagnosis, the site-specific mortality rates mirror the incidence rate. However, recent advances in the early detection and treatment of colorectal cancer (CRC) have significantly improved its prognosis. Geographical and temporal trends are the key factors influencing the variation in the incidence and mortality rate of GIC [14]. These demographic details demonstrate that GIC is a major health issue of our society. A better understanding of the underlying molecular mechanisms is required to develop effective therapeutic interventions for GIC patients.

In order to identify the origin of GIC, cell lineage tracking studies in transgenic mice have identified the cells expressing markers of stemness as the source of intestinal cancer [15,16]. Additional molecular studies have established a crucial role of inflammatory molecules in the malignant transformation of ISCs [17]. Further, next-generation sequencing studies have identified molecular similarities between ISCs and malignant cells of the GI tract [15,18]. These observations confirm that stem cells are the cell-of-origin of intestinal cancer. Within the tumor, cellular heterogeneity expands as the tumor grows. Among different types of malignant cells in an advanced tumor, cancer cells with stem-like properties [cancer stem cells (CSCs)] contribute significantly to metastatic spread, drug resistance, and tumor recurrence [19,20,21].

The CSCs possess various tumorigenic properties, including enhanced migration, invasion, and intravasation [22]. Additionally, CSCs exhibit resistance to apoptosis allowing them to survive in circulation and colonize at distant sites [23]. Moreover, CSCs interact with the cells of the tumor microenvironment (TME), including stromal and immune cells, and upregulate several transcription factors such as Oct4, Sox2, Nanog, and c-Myc, that have pro-cancer stem cell activity in GICs. Further, these interactions activate various signaling pathways, such as Wnt/β-catenin, the neurogenic locus notch homolog protein pathway (Notch), and Hedgehog, to promote CSC maintenance and expansion in GIC [24]. These interactions create a supportive niche for CSCs, promoting their survival, invasion, and metastatic colonization. Since CSCs significantly contribute to the initiation, progression, and metastasis of GICs, targeting their transcription factors and associated pathways holds promise for developing novel therapeutic strategies against GICs. In addition, many studies have evaluated the potential of CSCs and their molecular markers in the diagnosis and prognosis of GICs. Next-generation sequencing and multiplex immunostaining techniques are being used in discovery phase studies to develop CSC-based next-generation biomarkers for GICs. Several studies have also investigated the potential of CSCs as novel therapeutic targets and many ongoing clinical trials are evaluating their therapeutic potential. In this review, we summarize the role of inflammation and stem cells in the development and homeostasis of the GI tract. We further describe the role of CSCs and inflammatory molecules in the initiation, progression, and metastasis of GIC. We have also reviewed studies evaluating the potential of CSCs and their molecular markers in the diagnosis and treatment of GICs.

## 2. Stem Cells in the Development of the Gut

The presence of stem cells in the growing GI tract confers the ability to generate a variety of cell types within the GI tract and also provides a rejuvenating capacity to repair the damages. During embryonic development, the endoderm leads to the formation of the annulus region of villi, which consists of +4 label-retaining cells (+4LRC) stem cells, Leu-rich repeat-containing G protein-coupled receptor 5-expressing crypt base columnar cells (Lgr5+ CBC) stem cells, and Paneth cells. The base of the villi region (crypt region) is made up of Lgr5+ CBC stem cells and Paneth cells. The villi region above the crypt is made up of +4LRC stem cells, enteroendocrine, absorptive progenitor, and secretory progenitor cells. This annulus region is surrounded by mesenchymal cells. Epithelial–mesenchymal interactions induce the evaginations of villi in the small intestine, and after birth, these interactions further contribute to the invaginations creating the crypt region of the villi [25,26]. The crypt mainly comprises Lgr5+ CBC cells and Paneth cells [27]. The crypt CBC stem cells possess stemness properties and contribute to homeostasis. Lgr5+ CBC cells have an Lgr5 Wnt responsive gene, indicating that the expression of the Lgr5+ receptor enables CBC to respond to signals originating from mesenchyme in the base region. Evidence has shown that in the stomach and hair follicles, stem cells are found in the neck and bulge areas, rather than near the base of the epithelial invaginations [28,29,30,31,32,33]. Thus, it is possible that the traditional locations, which were initially identified using long-term label-retaining assays, indicate the presence of stem cells that remain in a prolonged inactive or quiescent state due to their inhibitory microenvironment. On the other hand, the Lgr5+ cells represent a population of stem cells that are sensitive to stimulating signals produced by surrounding mesenchymal cells, such as BMP antagonists like Noggin or Gremlin [34,35]. Nevertheless, additional research is required to ascertain whether this division of stem cells into sub-compartments is maintained in the intestine. For example, selective ablation of Lgr5+ CBCs will highlight their significant and unique role, and determine if +4 LRCs may generate CBCs. In contrast, targeted elimination of +4 label-retaining cells (LRCs) using a marker that particularly co-localizes with this cell type would indicate its exclusive status and reveal whether CBCs can also serve as replacements for lost +4 LRCs. The quiescent +4LRC cells construct the villi region above the crypt area. Although + 4 LRC cells do not perform self-replication, they undergo transient activation to generate CBC under stress conditions. Ultimately, it may be necessary to ablate both cell types to confirm and establish their mutual identification as intestinal stem cells (ISCs) [27,36]. Finally, future experiments, including live imaging, may be able to demonstrate cell migration and lineage. The regulation of these morphogenetic processes throughout the entire tract is poorly understood. However, it seems that Wnt, Notch, BMP, and HH signals play an important role. Simultaneously, epithelial cells within the lining of the GI tract undergo differentiation into specific subtypes.

The Lgr5+ and +4LRC cells contribute not only to GI development but to other organs as well. For instance, they can be found in the hair follicle, hematopoietic system, and skeletal muscle tissue [37,38].

The secretory lineage of the small intestine epithelium is made up of goblet cells, tuft cells, enteroendocrine cells, and Paneth cells. Goblet cells produce mucins that form a thick mucus barrier, protecting the epithelium from the contents of the lumen and microbiota [39]. Simultaneously, the mucus layer can serve as a nutritional source for commensal bacterial species [40,41]. Tuft cells, which make up less than 0.4% of the epithelium, function as chemosensors in the gut, and use taste receptors to monitor the interior of the intestinal lumen [42,43]. In addition, Tuft cells can also be identified by their production and expression of doublecortin-like kinase 1 and play a role in protecting the host against parasitic infection by releasing IL 25, a crucial cytokine for eliminating parasites [44,45,46,47]. Enteroendocrine cells are another key element of the secretory lineage. These cells are dispersed throughout the intestinal epithelium and play a critical role in regulating numerous physiological processes by secreting hormones such as glucagon-like peptide-1 (GLP-1), peptide YY (PYY), and cholecystokinin (CCK). These hormones are involved in regulating appetite, insulin secretion, and digestive enzyme release, thereby linking the gut to metabolic control and energy homeostasis [48,49]. Paneth cells, located at the crypt base, are essential for maintaining gut health by secreting antimicrobial peptides, such as lysozyme and α-defensins, which safeguard the stem cell niche and regulate the microbial population within the intestine [50]. Additionally, Paneth cells support the maintenance of ISCs by secreting growth factors such as Wnt, which are essential for the proliferation and differentiation of stem cells [51]. The absorptive lineage of the small intestine epithelium primarily comprises enterocytes, specialized for nutrient absorption. These cells also contribute to the immune response by interacting with gut microbiota and producing antimicrobial peptides [52]. Furthermore, enterocytes contribute to the maintenance of intestinal homeostasis through their involvement in the transport of ions and water, ensuring proper fluid balance within the gut lumen [53].

Finally, the small intestine, comprising villi and crypts, is lined by a single layer of epithelial cells. This layer includes stem cells, transit-amplifying cells, and other types of differentiated subtypes of cells such as secreting Paneth cells, enteroendocrine cells, goblet cells, tuft cells, and absorbing enterocytes [54].

## 3. Signaling Pathway in Stem Cell Regulation

Although the precise underlying molecular mechanisms governing the morphogenetic process are not fully understood, evidence indicates that the Wnt and Hedgehog signaling pathways play critical roles. Early studies indicated that the active Wnt signaling in the intervillus regions inhibited differentiation while stimulating proliferation. Disruption of Wnt signaling through Tcf4 inhibition ceased endodermal proliferation and depleted endodermal stem/progenitor cells [55]. These experiments also revealed that proliferative cells in the intervillus region were replaced by mature enterocytes typically found in the villus. However, recent findings suggest a more complex role for Wnt signaling during intestinal development. Using TOP-GAL and Axin2LacZ transgenic mice, Wnt activity was observed exclusively in the villus epithelium from embryonic day 16 (E16) to postnatal day 2 (P2), with no activity in the intervillus region until P2. By embryonic day 14 (E14), the villus and intervillus regions displayed distinct genetic signatures, and by embryonic day 17.5 (E17.5), proliferative cells were confined to the intervillus regions. While nuclear β-catenin and c-Myc expression corresponded to Wnt-active villus cells, other Wnt target genes and pathway components such as CD44, Cyclin D1, and Tcf4 were present in the intervillus region [56]. These results imply that Wnt signaling and proliferation may be uncoupled during early development and that the early epithelial proliferation might be driven by Wnt-independent mechanisms [56]. Additionally, Wnt signaling appears to be crucial for initial villus formation. Indian hedgehog, the Hedgehog signaling ligand, is expressed in the intervillus region, whereas its receptor, Patched, is found in the adjacent mesoderm [57]. The inhibition of Hedgehog signaling impairs villus formation, suggesting its function as a morphogen in the intestine, similar to other tissues [58]. It is unclear whether Indian hedgehog signaling is dependent on or independent of Wnt activation. Hedgehog signaling regulates the expression of BMPs primarily in mesoderm-derived mesenchymal cells, and BMP signaling plays a role in intestinal morphogenesis. The inhibition of BMP signaling by the overexpression of its inhibitor, Noggin, or through the conditional inactivation of its receptor, BMPR1A, leads to ectopic crypt formation, indicating the role of BMP signaling in limiting the crypt numbers [34,59,60].

## 4. GI Damage and Its Repair by Stem Cells

The GI epithelium faces continuous exposure to a diverse range of potentially damaging factors and needs to be repaired over time by the body itself. Monosodium glutamate (MSG), a food flavor enhancer, may weaken the intestinal barrier by promoting colorectal cancer (CRC) cell proliferation. This involves upregulating growth-related genes, adenomatous polyposis coli (APC) and beclin (BECN), while downregulating the tumor suppressor gene TP53, potentially disrupting the barrier [61]. Selected food additives such as polysorbates (used in cakes as whipped toppings, and cake fillings), aluminosilicates (used as anticaking and thickening agents, as pharmaceutical excipients, and in toothpaste), titanium oxides (common whitening and brightening agents used in food industry and pharmaceutical industry), maltodextrin (MDX; thickening agent), carrageenan (gelling agent), and saccharin and aspartame (artificial sweeteners) can disrupt the mucosal barrier in the GI tract, which leads to the infiltration of microorganisms [62,63,64,65,66,67,68]. This disruption increases the production of pro-inflammatory cytokines such as TNF, IL-2, IL-10, IL-18, IL1- β, and NF-kB as well as ROS, resulting in inflammation. In essence, food additives have the potential to compromise the intestinal permeability and trigger an inflammatory response in the gut [69,70,71,72,73,74,75].

Various food allergens such as gluten, a protein found in wheat, barley, and rye, can provoke mild allergies or sensitivities in some populations of Europe and the Asia-Pacific region, leading to various inflammatory responses. In individuals with conditions such as celiac disease, and non-celiac gluten sensitivity (NCGS), gluten acts as a trigger for GI discomfort and systemic inflammation [76,77]. This inflammatory response involves the release of pro-inflammatory cytokines such as TNF-α, IL-6, and IFN-γ, which not only contribute to chronic inflammation but also compromise the integrity of the intestinal barrier, allowing harmful substances to enter the bloodstream [77,78]. In celiac disease, an autoimmune response to gluten leads to persistent inflammation in the small intestine, resulting in tissue damage that significantly increases the risk of GICs, such as small intestinal adenocarcinoma and enteropathy-associated T-cell lymphoma (EATL) [79].

GIC is considered an old-age disease. Recent studies have uncovered the existence of clonal hematopoiesis of indeterminate potential (CHIP), defined as the acquisition of somatic mutations, leading to their clonal expansion [80]. CHIP has been shown to induce inflammation by generating increased neutrophil extracellular tracks (NETs). A study explored the incidence of CHIP in circulatory tumor DNA from GICs and its association with clinical outcomes. The study reported a higher average age in CHIP-(positive) patients compared with CHIP-(negative)− patients. This study reported no significant difference in the progression-free survival (PFS) or overall survival (OS) between CHIP+ and CHIP-(negative) groups [81].

To repair and restore epithelial damage, the GI epithelium has developed a remarkable regenerative capacity [46]. When damage occurs, the intestinal epithelial cells neighboring the damaged area lose their columnar polarity and migrate to cover the wound, a phenomenon known as “epithelial restitution” [82]. The process of epithelial restitution is independent of cellular proliferation and is regulated by inflammatory molecules like TGF-α, EGF, IL-1 β, and (IFN)-γ by inducing the production of bioactive transforming growth factor (TGF-β1) within epithelial cells [83]. After epithelial restitution, subsequent stages of wound healing are initiated, including enhanced epithelial cell proliferation and differentiation. These sequential steps occur within hours and days after the initial inflammatory damage [84,85]. Cell proliferation is crucial to replenish the pool of intestinal epithelial cells (IECs) needed to repair the tissue damage. These cells receive signals from inflammatory molecules, including IL-6 and IL-12, and Toll-like receptor (TLR) ligands like LPS and CpG DNA, that activate master transcription factors NF-kB and STAT-3, thereby promoting cell survival and proliferation [86,87,88]. In addition, it has been shown that TLR2 can suppress the apoptosis of IECs, and this outcome is achieved by the selective regulation of trefoil factor 3 (TFF) expression. Moreover, TLR2 plays a role in controlling the process of intestinal epithelial wound repair by altering the expression of epithelial connexin-43 (Cx43) [89,90]. Finally, differentiation and maturation are needed for re-establishing and maintaining the normal mucosal barrier functions. Under normal physiological conditions, the Lgr5+ ISCs, located at the base of the crypts, undergo a series of differentiation steps. These ISCs differentiate into transient amplifying progenitors that display a finite lifespan, which, in turn, further specialize into absorptive (enterocyte) and secretory progenitors under the control of Wnt/Notch signaling [91,92]. Secretory precursors can either develop into enteroendocrine cells through a Neurog3-dependent mechanism or can differentiate into Goblet or Paneth cells upon the activation of Atoh1 (also known as Math1). Later on, different cell types acquire lineage-specific expression of transcription factors (TFs), including Sox9 for Paneth cells and Klf4 for Goblet cells [93,94]. Taken together, this orchestrated differentiation and maturation process of ISCs plays a critical role in facilitating the recovery and regeneration of the GI epithelium from injury and inflammatory damage and maintaining homeostasis [95]. Within the intricate microenvironment of the GI tract lies a crucial determinant of mucosal homeostasis—the stem cell niche. This peculiar microenvironment provides vital signals and cues essential for the maintenance, regulation, and functionality of ISCs [96]. Additionally, Paneth cells play a crucial role in the immune defense of the mucosal lining and the maintenance of stem cell function by secreting growth factors, antimicrobial peptides, and Wnt ligands. These cells serve an integral role as niche cells by releasing essential growth factors such as EGF, TGF-α, Wnt3, and the Notch ligand, DLL4. Each of these factors triggers vital signals essential for maintaining and regulating stem cell activity [97].

In addition to epithelial cells, inflammatory molecules secreted by various types of stromal cells also contribute to the self-renewal and differentiation of ISCs. Furthermore, immune cells tend to reside in lamina propria just beneath the crypts [98]. Non-epithelial cells of the intestine, including fibroblast and endothelial cells, and various types of immune cells, create ISC niches and orchestrate the complex process of ISC self-renewal and differentiation [99]. Immune cells regulate and control ISCs by producing cytokines and other factors, essential for maintaining the integrity of the intestinal barrier, which is crucial in preventing both intestinal and systemic disease [100]. For example, type 3 innate lymphoid cells (ILC3s) secrete IL-22, which encourages the regeneration of ISC-mediated epithelial cells after intestinal damage and also drives ISC self-renewal [101]. Furthermore, ILC2s via IL-13 pathway promote ISC self-renewal [102]. Immune cells produce cytokines that also direct the development of Lgr5^high^ ISCs. A recent study has shown the control of the expression of Atoh1, a key element in lineage specification in the Notch in Lgr^high^ ISCs, thus inducing the differentiation of the secretory cell type. Figure 1 provides a summary of stem cell-GI tract homeostasis.

## 5. Inflammation, Stem Cells, Cancer Initiation, and Progression

The bidirectional connection between inflammation and stem cells has a direct impact on cancer development [103]. Chronic GI inflammation plays a significant role by inducing the production of reactive oxygen species (ROS) and promoting DNA damage, elevating the risk of mutations and cellular transformation [104]. Mutations in TP53, KRAS, SMAD, and APC genes disrupt the essential cellular processes and contribute to GIC initiation, development, and progression [105]. Multiple studies in transgenic mouse models have shown the ISC-specific origin of GIC [15,16,18,106]. APC gene deletion in LGR5+ stem cells leads to their transformation into malignant cells [106]. Similarly, oncogenic targeting of SOX9+ and SOX2+ stem cells induces malignant transformation [16]. The upregulation of OCT4 has been demonstrated to play a role in promoting the migration of tumor cells and their resistance to cancer therapeutics [107]. Chronic inflammation can activate a variety of signaling pathways such as Wnt, Notch, and Hedgehog, which are essential for regulating stem cell behavior [108,109,110]. The aberrant activation of these pathways can result in the over-proliferation of stem cells and the accumulation of genetic mutation, raising the risk of malignant transformation [111]. The inflammatory molecule TNF-alpha activates NF-kB and Wnt/beta-catenin pathways in the ISCs which transform them into malignant cells [15,18]. Inflammatory signals in the GI tract create a microenvironment that disrupts the delicate balance between renewal and differentiation of stem cells [112]. Inflammatory mediators appear to have a crucial role in inducing the expression of genes related to stemness. The expression of genes associated with stemness appears to be linked to the formation and evolution of cell compartments that are capable of regenerating CSCs [113,114].

A variety of oncogenic signals, including oncogenic activation, ROS, and inflammation, are known to activate cellular senescence, which refers to a permanently irreversible state where cells stop dividing, typically in response to various cellular stresses [115]. These cells secrete various factors known as the senescence-associated secretory phenotype (SASP), which contribute to cancer progression through different mechanisms, including ROS production, genomic instability, and a pro-tumor microenvironment [115,116]. It has been shown that the SASP factors, including IL6 and IL8, promote EMT, invasion, and stemness induction in cancers including GICs [117,118,119,120,121,122]. Additionally, the inflammatory signals produced by SASP, including IL-6, have been shown to influence cancer stem cells (CSCs) by activating pathways like STAT3 and NF-κB, which are crucial for maintaining the stemness and self-renewal capabilities of CSCs [123]. CSCs play a critical role in resistance to chemotherapy. Studies suggest that tumor-induced stress activates senescence. Many other studies have shown that chemotherapy treatment is similar to stress and leads to senescence in cancer cells. Mechanistic studies have shown that the chemotherapy doxorubicin-induced SASP promotes the expression of stemness-related genes, such as epithelial cell adhesion molecule (EpCAM), cytoskeletal 19 (CK19), annexin A3 (ANXA3), and the multidrug-resistance-related gene ABCG2 in GICs [124].

Recent studies have highlighted the functional role of CSCs in promoting the aggressive behavior and metastasis of GICs [125,126]. CSCs exhibit the capacity for self-renewal and the potential to differentiate into distinct cell types; similar to the dynamics seen in the normal intestinal epithelium in ulcerative colitis patients (UC), ALDH^high^ Wnt^high^ has been found as a marker panel for precancerous colonic stem cells (pCCSC), suggesting an elevated risk of malignant transformation. ALDH, expressed together with high levels of nuclear/cytoplasmic β-catenin, acts as a biomarker for pCCSCs and colorectal CSCs (CCSCs). Cells with high ALDH and WNT activity retain the ability to maintain tumorigenicity and tumor heterogeneity, underscoring their role in the transition from colitis to cancer [127]. An inflammatory environment promotes the enrichment of CSCs. Using the inflammation-induced colitis model, it was shown that the inflammation-induced Mucin 1, MUC1-C, promotes tumor progression by activating stemness pathways [128]. Mechanistically, MUC13 functions by protecting β-catenin from degradation, forming interactions with GSK-3β, which allows the movement of β-catenin into the nucleus and promote signaling, drive cancer initiation, progression, and invasion, as well as immune suppression [129]. Inhibiting the MUC1-C protein in colorectal cancer (CRC) cells reduces their capacity for wound healing, tissue invasion, self-renewal, and tumor formation. This indicates that MUC1-C may be a promising target for developing therapies to treat colitis and can hinder the progression of CRC. Endoplasmic reticulum (ER) stress is also known to induce inflammation [130]. One of the ER-stress-inducing genes, Tribbles homolog 3 (TRIB3), has emerged as a critical factor in promoting CRC stem cell properties. Mechanistically, Tribbles homolog 3 functions by recruiting β-catenin and TCF4, enhancing the transcriptional activity of this complex and activating genes that create a positive feedback loop, promoting CRC initiation and progression [131]. The ectopic expression of an actin-binding protein, Transgelin (TAGLN), promotes CRC and cholangiocarcinoma by enhancing stem cell programs [132,133]. Mechanistic studies revealed that TAGLN promotes cell proliferation, migration, and tumor formation by regulating the cytoskeleton and activates the oncogenic p38 MAPK pathway in CRC and cholangiocarcinoma [132,133]. Recent studies comparing normal, low-grade intraepithelial neoplasia (LGIN) in early gastric carcinoma using single-cell analysis revealed that the increased expression of DOT1 like H3K79 methyltransferase (DOT1L), a histone methyltransferase, an enzyme, triggers the activation of the Wnt/β-catenin pathway. This activation disrupts the destruction complex, leading to β-catenin stabilization and nuclear translocation, where β-catenin interacts with TCF/LEF transcription factors to activate target genes, including CD44, promoting CSC-like properties in gastric cancer cells [134]. While inflammation is proposed as one of the stimuli in initiating the transcriptional changes leading to the formation of CSCs, once CSCs are formed, they can contribute to the further amplification of inflammatory signaling. CSCs, which are also chemoresistant, have been shown to express proinflammatory gene signatures, mostly as a result of the continuous activation of NF-kB and interferon-stimulated regulatory element (ISRE)-dependent pathways. Inflammatory environments notably have tumor-associated macrophages (TAMs), which play a significant role in protecting tumor cells from chemotherapy treatment by facilitating and elevating the tumor growth properties of CSCs [135]. Figure 2 summarizes the signaling pathways of CSC activation.

## 6. CSC-Mediated Drug Resistance in Gastrointestinal Cancer

Chemotherapy is the primary treatment approach for GICs, and the overall survival rate has improved with the development of new drugs and treatment protocols. Nonetheless, GICs still experience high recurrence and metastasis rates, which are closely linked to drug resistance to cancer therapy. Numerous studies have shown that CSCs, which can arise from a small number of drug-resistant cells that survive chemotherapy, possess significant tumorigenic and self-renewal abilities. CSCs typically exhibit strong drug resistance and can serve as seeds for tumor recurrence and metastasis [136]. Recently, various surface markers such as ALDH, CD133, CD13, CD24, and CD44 have been identified in stem cells [137]; the high expression of these specific biomarkers has been closely associated with an increased likelihood of tumor drug resistance. In addition to this, the high expression of ATP2 binding cassette transporters (ABC transporters) in CSCs is a key factor for GIC drug resistance [138]. ABC transporters are known to be involved in the drug efflux from the cells, lowering the intracellular concentration of the drug, thereby reducing its effectiveness. This phenomenon, known as multidrug resistance (MDR), can have a major impact on the efficacy of chemotherapy. The presence of ABC transporters is vital in various organisms (prokaryotes and eukaryotes) that share a conserved sequence region responsible for ATP binding [21]. ABC transporters actively use ATP hydrolysis for transporting the molecules across biological membranes, which involves ATP-driven conformational changes in the transporter protein of bacteria that allow the import of substrate in a unidirectional manner [139]. Over 40 human ABC transporters have been discovered, with the family divided into seven different subfamilies (ABCA to ABCG). Several studies have indicated that ABC transporters can be employed as CSC surface markers for enrichment. The common sub-members of the ABC transporter family involved in the drug resistance, survival, and growth of CSCs are ABCB1, ABCB5, ABCC1, and ABCG2. ABC transporters on CSCs differ by cancer type, stage, and patient characteristics [140,141].

In a variety of tumor types, including GICs, a high expression of ABC transporter genes (mainly including MDR1/P-gp, MRP1) can expel chemotherapy drugs out of cancer cells, leading to drug resistance, thus accelerating cell survival and tumor recurrence [142]. Additional studies confirmed the role of ABC transporters in the development of drug resistance by promoting drug efflux in CSCs [143]. In terms of mechanistic studies, it has been found that ABC transporters affect tumor drug resistance through multiple signaling pathways, such as PI3K/Akt, NF-κB, Nrf2, Wnt/β-catenin, and JNK [144,145,146,147,148]. Some of these transporters, including P-glycoprotein, MRD-associated proteins, and breast cancer resistance protein, are involved in MDR development. These transporters are expressed in various tissues where they regulate the uptake, distribution, and elimination of drugs [149]. Additionally, these transporters influence the pharmacokinetics of drugs and xenobiotics [150].

Cancer-associated fibroblasts (CAFs) are known to produce and secrete IL-8, which triggers the activation of the NF-κB pathway. This activation subsequently leads to the upregulation of ABCB1 expression, an ABC transporter responsible for the efflux of drugs and the development of drug resistance. In colorectal cancer, this process results in a decrease in the intracellular accumulation of chemotherapy drugs such as doxorubicin, thereby diminishing the treatment efficacy [151]. Moreover, FOXO3 promotes proliferation and doxorubicin resistance in CRC cells by directly enhancing the expression of MDR1, another critical gene involved in drug efflux and resistance. This further contributes to a decrease in drug accumulation and the development of resistance to doxorubicin. In addition to IL-8 and FOXO3, the overexpression of Siva 1 in CRC cells results in reduced sensitivity to various anticancer drugs, such as vincristine, 5-fluorouracil, and doxorubicin [152]. Siva 1 has been shown to inhibit apoptosis, stimulate cell proliferation and migration, and enhance the expression of NF-κB, MDR1, and MRP1. These effects contribute to drug resistance by promoting cell survival, drug efflux, and reduced intracellular drug accumulation [153]. Furthermore, USP22 is involved in imparting resistance to oxaliplatin in CRC cells. The potential impact of USP22 on drug transport proteins, such as P-glycoprotein (P-gp) and MRP1, leads to reduced drug accumulation inside the cells and subsequent reduction in the drug efficacy [154]. This contributes to the development of resistance to oxaliplatin and presents a significant challenge in the therapeutic management of CRC. Collectively, these findings demonstrate the complex and diverse nature of drug resistance in GICs. Multiple factors, such as inflammatory cytokines, transcription factors like FOXO3, and proteins like Siva 1 and USP22, interact to promote drug resistance through their influence on drug efflux, diminished drug accumulation, augmented cell survival, and reduced drug sensitivity.

MicroRNAs (miRNAs) have emerged as crucial players in the modulation of drug resistance by targeting multiple genes and pathways. Several miRNAs have been identified to have significant roles in minimizing drug resistance and improving chemotherapy efficiency in GICs. The upregulation of miR-29a has been demonstrated to enhance the susceptibility of chemoresistant colon cancer cells towards doxorubicin (DOX) treatment. This doxorubicin susceptibility is achieved via the specific targeting of P-gp, phosphatase, and tensin homolog (PTEN) by miR-29a. Through the modulation of P-gp activity and the upregulation of PTEN expression, miR-29a exerts its inhibitory effects on drug resistance, facilitates apoptosis induction, and inhibits cell proliferation via the PI3K/Akt signaling pathway [155]. Similarly, it has been observed that miR-20b-5p plays a pivotal role in regulating drug resistance in CRC cells by targeting syndecan 2 (SDC2). SDC2 overexpression promotes drug resistance, cell survival, and invasiveness in 5-FU-resistant CRC cells. Conversely, the expression of miR-20b-5p exerts a negative regulatory effect on SDC2, which holds the potential to counteract drug resistance and augment the sensitivity of cancer cells to chemotherapy [156]. In CRC cells, the miR-506 has been shown to counteract resistance to the chemotherapy drug, oxaliplatin, by targeting the MDR1/P-gp drug efflux pump, which is responsible for drug efflux from the cancer cells, and concurrently inhibiting the Wnt/β-catenin pathway [157]. This leads to increased sensitivity of the chemoresistant cells to oxaliplatin, thereby enhancing the drug’s effectiveness. In gastric cancer cell lines, miR-200c negatively regulates the ABCB1 gene, which encodes P-gp. Through direct interaction with the messenger RNA (mRNA) of ABCB1, miR-200c exerts its influence by downregulating its expression, thereby leading to a decrease in P-gp levels within the malignant cells. The overexpression of miR-200c increases the sensitivity of drug-resistant cells to vincristine, an anticancer drug [158]. Cumulatively, these findings highlight the critical role of miRNAs in regulating drug resistance in GICs. miRNAs, such as miR-29a, miR-20b-5p, miR-506, and miR-200c, target key genes and pathways involved in drug resistance. Consequently, these intricate mechanisms regulate the chemoresistance and may be involved in improving the therapeutic outcomes. Harnessing the inherent potential of miRNA-based strategies to surmount drug resistance in GICs may pave the way for establishing novel and tailored therapeutic approaches for patients with GIC. Table 1 summarizes the molecular mechanism involved in the development of chemotherapy resistance in GIC.

## 7. Stem Cell-Based Prognostic Marker for GICs

Considering the significant biological role of CSCs in GIC initiation and progression, it is rational to evaluate their prognostic and therapeutic utilities. However, the detection of CSC poses a challenge due to the molecular similarities between tumor cells and CSCs. Recent research has delineated the list of markers to identify CSC phenotype (Table 2). Different cell surface markers, such as CD133, CD44, CD166, CD24, CD54, and CD90, have been recognized as indicators of GICSCs [159,160,161,162,163,164]. Additionally, other surface biomarkers such as CXCR4, EpCAM, LGR5, and LINGO2 also play crucial roles in identifying CSCs. Furthermore, pluripotent factors or intracellular stem cell-related factors and molecular pathways, i.e., aldehyde dehydrogenase (ALDH), leucine zipper/EF-hand-containing transmembrane protein 1 (LETM1), Mushashi2, Nanog, octamer-binding transcription factor 3/4 (Oct-3/4), sex-determining Y-box 2 (SOX2), Sall4, Notch, Wnt/beta-catenin, and alpha-fetoprotein (AFP) serve as crucial regulators of GI CSCs [16,107,165,166,167,168,169,170,171,172,173,174,175,176,177,178,179,180,181,182,183,184,185,186,187,188,189,190,191,192]. We have summarized diagnostic and prognostic markers in Table 2 and Table 3 respectively.

Recent findings have highlighted the relationship between the increased expression of key CSC-related markers and certain signaling pathways, which hold both prognostic and clinical significance in GIC. However, there are ongoing conflicts as a result of differences in methodology, the number of participants, and the diversity of the study participants. Hence, there is a pressing need to clarify this relationship through meta-analysis and clinical data to identify new biomarkers [199]. The section below summarizes recent studies examining CSC-based prognostic markers in GIC.

Analysis of MUC1-C expression in colitis-associated CRC tissue samples showed that MUC1-C inversely correlates with patient survival. Another study using colon cancer tissue array reported MUC13 as a poor prognostic marker [129]. Remarkably, the researchers also discovered a 20-gene signature that has been dysregulated in peritoneal metastasis. These genes are linked to the concept of “stemness”, suggesting a connection between CSCs and changes in the inflammatory environment that benefit tumors. Importantly, this gene signature is not just informative; it can predict how well a patient might do in terms of surviving cancer (overall survival) and remaining disease-free (disease-free survival). These findings indicate the possibility of using this gene signature in precision medicine strategies specifically tailored for cases of peritoneal carcinomatosis [200].

Another well-known stem cell marker, SOX9, has been identified as a marker to predict disease recurrence in patients with gastric cancer (GC) [16,201]. This observation was derived from immunohistochemical analysis of multiple and large cohorts of patient samples. In addition, aquaporins (AQP), which are dysregulated in various inflammatory diseases, are also dysregulated in GC [202,203]. Concurrently, AQP5 is expressed in primary tumors and metastases of intestinal and diffuse GC subtypes. AQP5+ tumor-resident cells exhibit stem potential ex vivo, suggesting AQP5 as a biomarker for diagnosis, prognosis, and therapeutic targeting. AQP5 is identified as a specific marker of GC-CSCs, coexpressed with LGR5, promoting tumorigenesis. Mechanistically, AQP5 activates autophagy by inducing LC3I/LC3II transformation in GC-CSCs, crucial for AQP5’s functions, and recruits E3 ligase TRIM21 to ULK1, inducing K63-mediated ubiquitination of ULK1 [198]. The expression of actin-binding protein TAGLN correlates with advanced CRC stages and poor overall survival, suggesting TAGLN as a potential prognostic marker for advanced CRC stages. However, a correlation of TAGLN with cholangiocarcinoma prognosis was not observed [114,115]. Collectively, these studies highlight the critical role of several biomarkers and pathways in GIC advancement and emphasize potential approaches for targeted therapies and improved prognostic strategies [204]. We have summarized diagnostic and prognostic biomarkers in Table 2 and Table 3.

## 8. Cancer Stem Cell-Based Targeted Therapies and Clinical Trials

The most defining characteristic of CSCs is their ability to continuously self-renew and sustain multiple tumor differentiation. In normal stem cells, self-renewal and differentiation are tightly controlled by signaling pathways such as Wnt/β-catenin, (Notch), Hedgehog, and BMP pathways, which are notably dysregulated in CSCs. These signaling pathways function not as isolated regulators but as a complex network that collectively governs CSC stemness, granting them the potential for self-renewal, proliferation, multidirectional differentiation, and the capacity to initiate tumors, reconstruct tumors, and resist radiotherapy [205]. Traditional cancer treatments target proliferating and mature cancer cells; however, CSCs have led to the development of new therapeutic strategies focused on eliminating CSCs that drive tumor growth and reconstruction, rather than simply reducing tumor size [206]. The mechanism upholding the self-renewal characteristics of CSCs coincides with the fundamentally critical pathways necessary for the identification and development of anti-cancer drugs that specifically target CSCs [207]. As mentioned, the disruption or increased activation of Wnt/beta-catenin, Notch, Hedgehog, and BMP signaling pathways may extensively contribute to the CSC recurrence as well as its persistence [208,209,210]. Nevertheless, these signaling pathways also play a crucial role in the regulation of normal functioning of stem cells. Therefore, it becomes imperative to develop therapies that will target CSCs, while minimizing any potential adverse consequences resulting from interfering with the normal activity of stem cells.

### Clinical Trials

To evaluate the clinical potential of targeting CSC-activating pathways against GICs, several clinical trials have examined different pharmacological inhibitors and monoclonal antibodies targeting different components of CSC-activating pathways. We search relevant clinical trials using ClinicalTrial.gov and Pubmed databases. The PRISMA FLOW Chart 2020 describes the clinical trial search strategy (Figure 3). Since Wnt signaling has an important role in the activation of GICSCs, many clinical trials have evaluated the clinical potential of different Wnt pathway inhibitors in GIC. In this context, the inhibition of the Dickkopf-1 (DKK1) molecule by the DKN-01 monoclonal antibody is being evaluated in several clinical trials [211]. DKK1 is a Wnt pathway activator. A phase 2 study (NCT04363801) targeting gastric or gastroesophageal (GEJ) cancer involving 25 patients delved into DKN-01/Tislelizumab ± chemotherapy, an inhibitor of the Wnt pathway, currently recruiting to delineate its effects on CSCs. Two years of follow-up data suggested an improved median progression-free and median overall survival (mPFS: 11.3 vs. 7.7 months; mOS: 19.5 vs. 13.8 months) with a manageable safety profile compared with Tiselizumab ± chemotherapy. Interestingly, the mPFS and mOS were shorter in patients with lower PD-L1 expression. These findings suggest that the GEJ adenocarcinoma and GC patients with DKK1 and PD-L1 expression might be treated with the combination of DKN-01/Tislelizumab ± chemotherapy. In addition, future studies describing the mode of action will facilitate the development of effective immunotherapies. In metastatic CRC (NCT02278133), a completed phase 1b/2 trial with 20 patients explored Wnt pathway inhibition with a combination of Wnt pathway inhibitors WNT974, LGX818, and Cetuximab. However, this treatment led to bone toxicities. Serious adverse events were reported in 15 patients. Therefore, the clinical trial was discontinued and only 10% overall response was reported [212].

Hedgehog pathways play a critical role in the maintenance of CSCs [213]. The phase 2 trial exploring the inhibitor of smoothened (SMO) (vismodegib, GDC-0449) in metastatic colon cancer (NCT00636610) involved 98 patients treated with Vismodegib and 101 patients with placebo to Vismodegib, and the Hedgehog pathway was targeted using an SMO inhibitor. The study revealed a progression-free survival (PFS) of 9.3 months with Vismodegib and 10.1 months with Placebo to Vismodegib (with a 90% confidence interval). This observation suggests that vismodegib has limited efficacy in these patients. One of the explanations for this observation is that the inability of vismodegib to inhibit metastatic colon cancer is probably due to the mutations in the SMO. SMO mutations have been shown to prevent vismodegib activity in basal cell carcinoma [214,215,216].

The Notch pathway is also known to maintain CSCs. The Notch inhibitor, CB-103, efficiently inhibits the interactions between CSL-NICD, which is activated by the Notch pathway [217]. The phase 1/2 study (NCT03422679) exploring the therapeutic potential of CB-103 showed that it stabilizes disease progression. Another phase 2 study evaluating the utility of a Notch inhibitor, RO4929097, reported no clinical benefit in metastatic CRC (NCT01116687). However, the molecular mechanism explaining the lack of RO4929097 efficacy is not clear. A possible mechanism is the autoinduction of RO4929097 metabolism. The gamma-secretase inhibitor as a single therapy might not be efficient. Gamma-secretase inhibitors might be efficient with other cytotoxic agents, as shown in a previous in-vitro study using CRC cell lines [218]. Although abrupt Notch signaling has a role in promoting GIC progression, the clinical trials evaluating the possibility of targeting the Notch1 pathway have not generated any promising results. We have summarized the clinical trials in Table 4.

**Figure 3 cancers-16-03134-f003:**
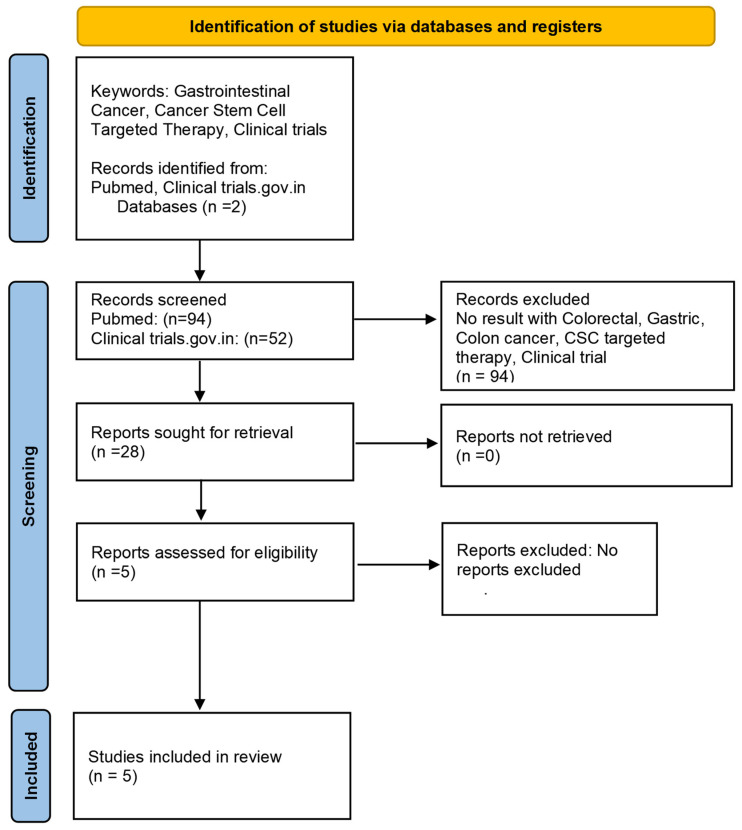
The flow chart describes the searches of databases, registries, and other sources utilized in this review. Created by (PRISMA FLOW Chart 2020) [219].

## 9. Conclusions

In conclusion, CSCs play a pivotal role in the progression of GICs, and their resistance to the treatment. Various mechanistic studies have defined the role of microRNAs, drug transporters, and inflammatory molecules. However, many known drug resistances observed in other cancers also needs to be explored in GI. For example, the role of drug-metabolizing enzymes, noncellular components, and epigenetic regulations. Although the relation of inflammation-induced stemness by activating senescence-related pathways has been explored, we still have very limited insights about the role of host cells on the activation of CSCs in GI. Understanding the underlying molecular mechanisms that drive CSC function and identifying distinct biomarkers are vital for the development of effective therapeutic approaches. Modern technologies have identified CSC-related molecules as promising prognostic biomarkers. Future studies exploring CSCs in patient’s blood may lead to the development of liquid biopsy-based prognostic biomarkers for follow-up and population screening protocols. Focusing on pathways and markers associated with CSCs presents a promising strategy to overcome drug resistance and enhance patient outcomes. This knowledge can further be harnessed to identify off-the-shelf drugs that might be repurposed as a combination therapy against GIC. As research progresses, incorporating CSC-targeted therapies into clinical practice may greatly improve the treatment of GICs and lower the risk of tumor recurrence. The failure of Notch1 inhibitor in human clinical trials suggests the requirement of rigorous preclinical studies. Similarly, the beneficial effects of DKN-01 on GICs should be confirmed in a larger cohort from multiple centers. In addition, the association of DKN-01 with immune pathway molecules might help in developing it as a predictive biomarker and a target for improved immunotherapy. This review highlights the necessity of further exploring CSC biology and developing innovative therapeutic approaches designed specifically to target these cells.

## Figures and Tables

**Figure 1 cancers-16-03134-f001:**
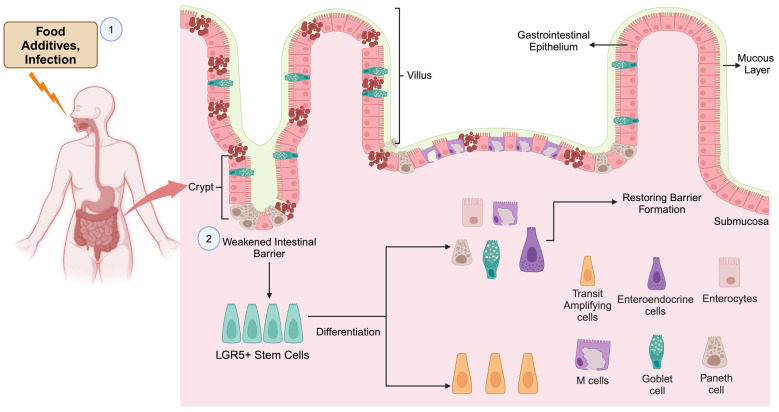
Stem cell-mediated repair of GI tract. (1) The organ-damaging agents, including food additives and infectious agents, induce intestinal damage. (2) Organ damage activates LRG5+ stem cells to produce different types of cells required for tissue repair. This original figure was created by biorender.com, accessed on 11 September 2024.

**Figure 2 cancers-16-03134-f002:**
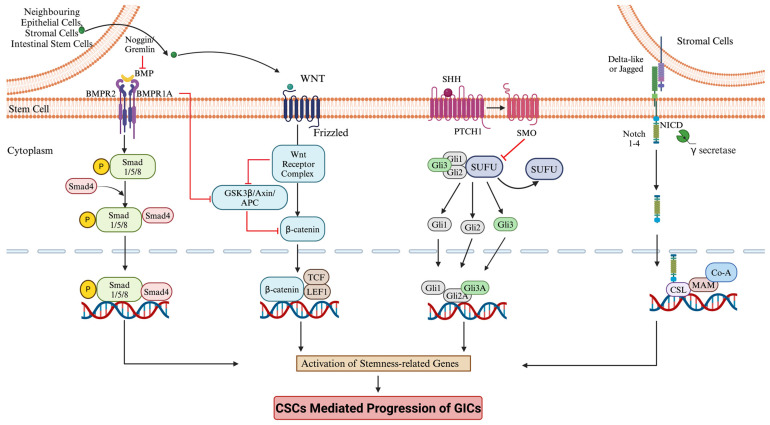
Signaling pathways of cancer stem cell activation. The inflammatory signals produced by the epithelial and stromal cells of the intestine activate various signaling pathways of stemness. Activation of stemness pathways enriches cancer stem cells that contribute to tumor progression. This original figure was created by biorender.com accessed on 11 September 2024.

**Table 1 cancers-16-03134-t001:** Molecular Mechanism of Chemotherapy Resistance in GIC.

Cancer Type	Chemotherapy	Pathway Involved in Resistance	Molecules Involved in Resistance	Reference
Colorectal Cancer	Doxorubicin	NF-kB	IL8, ABCB1	[151]
Colorectal Cancer	Doxorubicin	-	FOXO3, MDR1	[151]
Colorectal Cancer	Vincristine, 5-fluorouracil, doxorubicin	NF-kB	Siva 1, MDR1, MRP1	[152,153]
Colorectal Cancer	Oxaliplatin	-	USP22, P-glycoprotein, MRP1	[154]
Colon Cancer	Doxorubicin	PI3K/Akt	miR-29a, P-gp, PTEN	[155]
Colorectal Cancer	5-Fluorouracil	-	miR-20b-5p, SDC2	[156]
Colorectal Cancer	Oxaliplatin	Wnt/β-catenin	miR-506, MDR/P-gp	[157]
Gastric Cancer	Vincristine	-	miR-200c, ABCB1 (P-gp)	[158]

**Table 2 cancers-16-03134-t002:** Molecular Markers for the Detection of CSCs.

Surface CD Marker				
	Gastric CSCs	Reference	CRC CSCs	Reference
CD24	+	[162]	-	
CD44	+	[159]	+	[193]
CD90	+	[159]	-	
CD133	+	[194]	+	[193]
CD166	+	[195]	-	
Other Surface Biomarker				
EpCAM	+	[162]	+	[169]
LGR5	+		+	
Intracellular Markers				
Nanog	+		-	
Oct-3/4	-		-	
SOX2	+	[183]	-	
Sall4	-		+	[187]
AFP	-		-	

Note: (+) is presence of molecule, (-) is absence of molecule.

**Table 3 cancers-16-03134-t003:** Molecular Markers for Prognosis of CSCs.

Surface CD Biomarkers				
	Gastric CSCs	Reference	CRC CSCs	Reference
CD24	+	[162]	-	
CD44	+	[165,179]	+	[196]
CD54	+	[180]	-	
CD90	-		-	
CD133	+	[180]	+	[163]
CD166	+	[164]	-	
Other Surface Biomarkers				
CXCR4	+	[164]	-	
EpCAM	+	[197]	-	
LINGO2	+	[170]	-	
MUC13	-		+	[129]
DOTIL	+	[134]		
AQP5	+	[198]		
Intracellular Markers				
ALDH	+	[165,166,167]	+	[168]
TRIB3	-		+	
Letm1	+	[171]	+	[171]
Mushashi2	+	[173]	-	
Nanog	+	[174,175]	+	[178]
Oct-3/4	+	[180]	+	[182]
SOX2, SOX9	+	[16,174,175]	+	[186]
Sall4	-		+	[187]
MUC13	-	-	+	[129]
TAGLN	-		+	[114,115]

Note: (+) is presence of molecule, (-) is absence of molecule.

**Table 4 cancers-16-03134-t004:** List of Clinical Trials evaluating CSC in GICs.

Cancer Type	Inhibitor Reported	Targeted Pathway	Reported Outcomes	Clinical Trial Identifier
Gastric Cancer/Gastroesophageal Cancer	DKN-01, Tislelizumab	Wnt	Currently recruiting, outcomes not reported	NCT04363801
Metastatic Colorectal Cancer	WNT974, LGX818, Cetuximab	Wnt	Completed	NCT02278133
Metastatic Colon Cancer	Vismodegib	Hedgehog	PFS: 9.3 months (Vismodegib), 10.1 months (Placebo)	NCT00636610
Metastatic Colorectal Cancer	RO4929097 (Gamma-secretase inhibitor)	Notch	No benefit	NCT01116687
Colorectal Cancer	CB-103	Notch	PFS: 21.7 months	NCT03422679

## Data Availability

The data presented in this study are available in this article.

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
