# Peer review of "Inflammation-Associated Stem Cells in Gastrointestinal Cancers: Their Utility as Prognostic Biomarkers and Therapeutic Targets"

_cancers, 2024, doi:10.3390/cancers16183134_

Round 1
Reviewer 1 Report
Comments and Suggestions for Authors
The authors of this article reviewed current understanding of the biology and roles of stem cells during the development, homeostasis, and repair of the gastrointestinal tract (GI), focusing specifically on the influence of inflammation on modulating stem cells into cancer stem cells and contribute to cancer therapy resistance. This article provided a comprehensive view of stem cell and stemness-related chemotherapy resistance mechanisms, stem cell markers, and relevant targeted therapy clinical trials. The articles cited are appropriate, including seminal papers of key discoveries and more recent research articles.
Here are some suggestions and comments that may further strengthen this article:
While the title reads "GI cancers", the majority of efforts focused on the intestine (up until section 5) and gastric cancers are not equally highlighted. It is worth emphasizing the similarity and differences of these organs, current understanding, and respective gaps.
The 3rd paragraph in section 2 suffers from a lack of clarity in terminology and hypotheses referenced in Line 130. While the two stem cell populations in bone marrow is interesting, its connection to stem cell populations in the GI is not sufficiently discussed. Are there any evidence of other tissues having both quiescent and cycling stem cell populations?
Some topics were discussed in detail while related topics were only mentioned, such as in line 192-202: What about the other two types of cells, enteroendocrine and Paneth cells?
In section 3, the influence of food additives on GI epithelium damage were discussed. There are emerging evidence of common food ingredients that cause mild allergy/sensitivity in select populations, such as gluten in Caucasians, which may contribute to chronic inflammation, leaky gut, and cancer. However, this area is not discussed in this section, which may add more relevance to the article.
While this review may serve as a useful summary to stem cell-mediated chemo-resistance mechanisms and diagnostic markers etc, what are the key gaps in the CSC field in GI cancers? What questions do the authors' consider vital to further our understanding of inflammation-induced CSC in GI?
Comments on the Quality of English LanguageThe article is understandable and structured.
Several sentences lacks clarity, such as line 232-233, line 297-301, etc.
The overall flow of the article can be improved by rearranging paragraphs, such as Line 331-338 on the prevalence of GIC, to the very beginning of this review, to highlight the importance of understanding mechanisms of GIC. This would improve flow with the paragraphs before and after, and emphasize the importance of this review.
There are several duplicated sentences (e.g. line 278-281, line 340-342, etc) that repeats information already stated, and grammatical errors (e.g. line 384-385, etc).
Reviewer 2 Report
Comments and Suggestions for Authors
The manuscript describes a probable correlation between inflammation and cancer stem cell proliferation.
It is a relatively new aspect in looking at gastrointestinal cancer occurrence and progression, opening conceptual new therapeutic approaches.
Few suggestions: in the manuscript there are often excessive descriptions of histology and anatomy, which are well know to the reader. The manuscript should be shortened. For example the first lines of the introduction (lie 31-38) are a basic knowledge for the reads of Cancers and the introduction should be shortened.
Describing clinical trials, I suggest the Authors to describe the methods used to search and to select papers to be included (PRISMA FLOW Chart 2020).
I would suggest the Authors , if I am allowed to do so, to include a subchapter in the manuscript describing the role of clonal hematopoiesis of indeterminate potentials as possible correlation between inflammation and cancer (Libby P, Ebert BL. CHIP (Clonal Hematopoiesis of Indeterminate Potential). Circulation 2018;138: 666-668. doi: 10.1161/CIRCULATIONAHA.118.034392.
Comments on the Quality of English Language
Moderate editing of English language required.
Reviewer 3 Report
Comments and Suggestions for Authors
The review by Kumari et al. offers a comprehensive examination of CSCs in gastrointestinal cancers. It covers CSC characteristics, the role of various biomarkers and signaling pathways, and clinical trials targeting these pathways. The review effectively summarizes CSC biology and highlights several clinical trials aimed at targeting CSC pathways, including Wnt, Hedgehog, and Notch.
Major points:
However, the review has notable shortcomings. It lacks a focused narrative and presents information in a dense manner that is not engaging. Additionally, the review fails to critically evaluate key trial outcomes and their implications. For example, in the discussion of the Hedgehog pathway trials (page 14, line 577), it notes that a Phase II trial with Vismodegib reported a progression-free survival (PFS) of 9.3 months compared to 10.1 months with placebo. This suggests that Vismodegib might be less effective or that there were issues with the trial, but the review does not address this discrepancy or explore its potential reasons.
Similarly, the review mentions Notch pathway trials (page 14, line 590), including RO4929097 and CB-103, but does not critically assess the poor outcomes reported for RO4929097 (overall survival of 6 months, PFS of 1.8 months) or discuss the implications of the termination of the CB-103 trial. The review misses an opportunity to explore the reasons behind these outcomes and their impact on treatment strategies.
The conclusion emphasizes the importance of CSCs and the need for continued research but remains too broad. It lacks specific recommendations or actionable insights derived from the discussed trials. A more detailed discussion of the implications of trial results and future research directions would enhance its impact.
Moreover, there are current reviews with strong overlap, such as “Molecular Biomarkers and Signaling Pathways of Cancer Stem Cells in Colorectal Cancer” by Omran et al. in Technology in Cancer Research & Treatment (2024), and “Stem Cell Biomarkers and Tumorigenesis in Gastric Cancer” by Wuputra et al. in J Pers Med (2022).
Minor points:
Additional minor issues include the claim that no results are reported for trial NCT02358161, despite available insights (see https://ascopubs.org/doi/10.1200/JCO.2021.39.15_suppl.e16239), and an inconsistent focus on gastrointestinal cancers, while abrupt mentioning hepatocellular carcinoma and pancreatic cancer.
Spelling error: OCT-3/4 is misspelled (page 11, line 500).
Overall, the review has to be improved by focusing on unique findings, providing a more in-depth analysis of trial results, and offering specific recommendations for clinical practice.
Reviewer 4 Report
Comments and Suggestions for Authors
This review article regarding importance of inflammation-associated stem cells in developing GI tract cancers is interesting and well written. However, several points listed below should be reconsidered.
1) The title can be slightly modified based on the text described. I guess “Inflammation-associate stem cells” or “Inflammation-related stem cells” or “Inflammation-modified”, etc.
2) Recently, senescence-associated secretory phenotype (SASP) is known to lead to cancer progression and chemoresistance. Please mention the association among SASP, inflammation, stem cells, and GI cancer.
3) Please add representative immunohistochemical or immunofluorescence figures of stem cells in GI tract and GI tract cancers. They are helpful for readers who are interested in this theme.
4) I think too many references are listed. It is possible to cite only very important references.
Round 2
Reviewer 2 Report
Comments and Suggestions for Authors
Excellent review with important clinical implications
Reviewer 3 Report
Comments and Suggestions for Authors
The authors have notably improved the content and flow.
Reviewer 4 Report
Comments and Suggestions for Authors
The revised manuscript has been greatly improved.